# Squeeze-and-Excitation Encoder-Decoder Network for Kidney and Kidney Tumor Segmentation in CT images

Jianhui Wen[1], Zhaopei Li[1], Zhiqiang Shen[1], Yaoyong Zheng[1], and Shaohua Zheng[1,*]

College of Physics and Information Engineering, Fuzhou University, Fuzhou, China
`sunphen@fzu.edu.cn`

**Abstract.** Kidney cancer is one of the top ten cancers in the world, and its incidence is still increasing. Early detection and accurate treatment are the most effective control methods. The precise and automatic segmentation of kidney tumors in computed tomography (CT) is an important prerequisite for medical methods such as pathological localization and radiotherapy planning, However, due to the large differences in the shape, size, and location of kidney tumors, the accurate and automatic segmentation of kidney tumors still encounter great challenges. Recently, U-Net and its variants have been adopted to solve medical image segmentation problems. Although these methods achieved favorable performance, the long-range dependencies of feature maps learned by convolutional neural network (CNN) are overlooked, which leaves room for further improvement. In this paper, we propose an squeeze-and-excitation encoder-decoder network, named SeResUNet, for kidney and kidney tumor segmentation. SeResUNet is an U-Net-like architecture. The encoder of SeResUNet contains a SeResNet to learns high-level semantic features and model the long-range dependencies among different channels of the learned feature maps. The decoder is the same as the vanilla U-Net. The encoder and decoder are connected by the skip connections for feature concatenation. We used the kidney and kidney tumor segmentation 2021 dataset to evaluate the proposed method. The dice scores of SeResUNet in kidney, masses, and tumor are 91.6%, 58.8%, 54.16%, respectively.

**Keywords:** Kidney tumor segmentation · Squeeze-and-excitation network · U-Net.

## 1 Introduction

Kidney cancer is the malignant tumor with the highest mortality in the urinary system. Computed tomography (CT) imaging is the most common medical treatment for kidney cancer inspection and diagnosis. Segmenting kidneys and tumors from CT images is an important prerequisite for medical methods such as pathological localization and radiotherapy planning. This is usually done manually by professional medical personnel or staff with relevant backgrounds. However, manual segmentation of kidney and kidney tumor from a large number of

slices is time-consuming and suffers from human error. Automatic segmentation of the kidney and tumor can help doctors quickly locate the tumor and prepare for further surgical planning. The expansion of public databases has extremely promoted the segmentation of medical images. The kidney tumor segmentation challenge 2019 (KiTS19) [4] first released a public data set with kidney tumor annotations for the participants to develop automated segmentation approaches. KiTS19 provides 300 high-quality CT scan images of kidney cancer patients. Among them, 210 high-quality annotated CT scans are used for training, and 90 CT scans are used for algorithm testing. The KiTS19 Challenge has greatly promoted the segmentation of kidneys and kidney tumors.

Recently, deep learning-based methods have achieved impressive performance on medical image segmentation. Specifically, U-Net and its variants [11,1,14,8] are widely exploited for kidney and kidney tumor segmentation. For example, Yang et al. proposed a 3D full convolutional network combined with a pyramid pooling module (PPM) for kidney and kidney tumors segmentation, which can make full use of the 3D spatial contextual information to improve the segmentation of the kidney as well as the tumor lesion [12]. Abhinav Dhere et al. used the anatomical asymmetry of the kidney to define an effective kidney segmentation agent task through self-supervised learning [2]. Yu et al. proposed a framework named Crossbar-Net, which through vertical patches and horizontal patches to capture both the global and local appearance information of the kidney tumors, and cascade the horizontal sub-model with the vertical sub-model to segment the kidney and tumor [13]. Isensee et al. use the nnUnet for kidney and kidney tumor segmentation, which won the 1nd place in the kidney tumor segmentation challenge 2019 (KiTS2019) [7]. Hou et al. proposed a three-stage self-guided network to accurately segment kidney tumors. The first stage determines the rough position of the target, the second stage optimize, smooth kidney boundary and get the initial tumor segmentation result, the tumor refine net is proposed to optimize previous stage's tumor segmentation result in the third stage, which ranked the 2nd place in the KiTS19 [5]. Although these methods achieved favorable performance, the long-range dependencies of feature maps learned by convolutional neural network (CNN) are overlooked, which leaves room for further improvement.

Motivated by the squeeze-and-excitation network [6] to model long-range dependencies of the learned feature maps, in this paper, we propose a squeeze-and-excitation encoder-decoder network, named SeResUNet, for kidney and kidney tumor segmentation. Specifically, SeResUNet is an U-Net-like architecture including an encoder, a decoder, four skip connection paths. The encoder of SeResUNet contains a SeResNet to learns high-level semantic features and model the long-range dependencies among different channels of the learned feature maps. The decoder is the same as the vanilla U-Net. The encoder and decoder are connected by the skip connections for feature concatenation. We evaluated the proposed method on the 2021 kidney and kidney tumor segmentation challenge(KiTS21) dataset [4]. Experiment result shows that the dice scores of SeResUNet in kidney, masses, and tumor are 91.6%, 58.8%, 54.16%, respectively.

## 2    Method

In this section, we detail our architecture for automated kidney and kidney tumor segmentation in CT images. First, We introduced the overall architecture in Section 2.1. In Section 2.2, the squeeze-and-excitation module are specified. Then, in Section 2.3, we present the deep supervision used in this work. Finally, the loss function is discussed in Section 2.4.

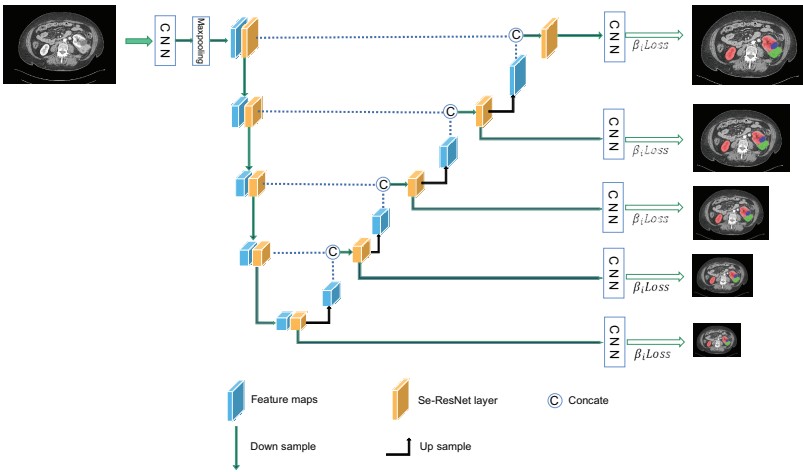

**Fig. 1.** Network architecture for segmentation

### 2.1    Architecture

The overview of our proposed framework is shown in Fig 1. Our method is an encoder-decoder architecture. The encoder adopts ResNet50 [3] as backbone, including four residual blocks followed by maxpooling layers, to gradually aggregate high-level semantic information. In addition, we exploit the Squeeze-and-Excitation(SE) module in the encoder to model long-range dependencies of channel relation among the input feature maps. Specifically, SE module transform the feature maps into a channel descriptor, then recalibrate the input features themselves by channel-wise multiplication. The decoder up-samples the high-level feature map to obtain the segmentation map with size the same as the original image. Each convolution layer of decoder is of kernel size of $3 \times 3$, stride of 1, and padding of 1. In order to avoid the problem of vanishing gradient and to train the proposed network quickly, we introduce multi-level deep supervision in the decoder, where deep supervision is performed on each layer of the decoder so that the shallow layer can be fully trained. After the four-layer up-sampling of the decoder, a segmentation map with the same size as the original image is obtained.

## 2.2   Squeeze-and-Excitation module

We employ Squeeze-and-Excitation(SE) module [6] to capture channel-dependencies of the learned features. The structure of the SE module is depicted in Fig 2.

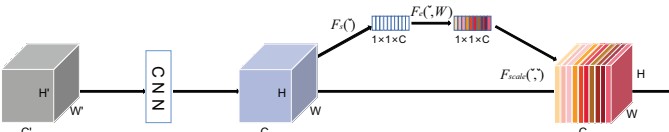

**Fig. 2.** Squeeze-and-Excitation module

**Squeeze** operation compresses each two-dimensional feature channel into a real number with a global receptive field, and the output dimension matches the input feature channel number. In short, it is to carry out global average pooling, the specific equation is as follows

$$S_c = F_s\left(\omega_c\right) = \frac{1}{H \times W} \sum_{\alpha=1}^{H} \sum_{\beta=1}^{W} \omega(\alpha, \beta) \tag{1}$$

where $\omega_c$ represents the $c_{th}$ feature map of size $H \times W$. After Eq. 1, the $H \times W \times C$ input is converted to $1 \times 1 \times C$, which represents the numerical distribution of the C feature maps in this layer, corresponding to the $F_s\left(\cdot\right)$ in Fig. 2.

**Excitation** is similar to the gate in the recurrent neural network, expresses the correlation between different feature channels by generating weights for each feature channel, the specific equation is as follows

$$E_c = F_e(S, W) = \sigma(g(S, W)) = \sigma\left(W_2 \delta\left(W_1 S\right)\right) \tag{2}$$

where $\sigma$ refers to the ReLU [9] function, $W_1, W_2$ is a fully connected layer with different parameters, used to fuse feature map information of different channels. The dimension of E obtained after Eq.2 is $1 \times 1 \times C$, where C is the number of channels.

**Recalibration** operation multiply the excitation output E by the previous features, completing the recalibration of the original feature in the channel dimension. The specific equation is as follows

$$F_{scale}\left(\omega_c, E_c\right) = \omega_c \cdot E_c \tag{3}$$

## 2.3   Deep supervision

To avoid the vanishing gradient problem and quickly train the proposed network, we perform deep supervision in the decoder, as shown in the right of

Fig.1. Specifically, each layer of the decoder predicts a segmentation map for the calculation of the loss function. This is different from multi-task learning (MTL). MTL has different ground truths to calculate different losses, while there is only one ground truth for deep supervision. Different network layers calculate loss and sum them according to different coefficients. The weighted coefficients are set as 0.4, 0.3, 0.2, 0.05, 0.05, respectively. Since the sizes of feature map of each output layer are different, we down-sample the ground truth to the same size of the corresponding output segmentation map for loss calculation.

### 2.4   Loss function

Loss function is used to estimate the degree of inconsistency between the predicted segmentation map $f(x)$ of the model and the ground truth $Y$. Considering the proportions of the volume of the kidney, tumor, cyst and background area are different, there is an imbalance in data distribution. Therefore, we use weighted cross-entropy (WCE) loss function to solve this problem.The specific definition is as follows

$$\mathcal{L}_{wce}(\beta, P) = \sum_{i=0}^{S} \beta^i P_{GT}^i \log \left(p_{pred}^i\right) \tag{4}$$

where S is the number of classes, specific for kidney, kidney tumor, kidney cyst and background. $P_{GT}^i$ and $p_{pred}^i$ are the probabilities of the $i_{th}$ class of the ground truth and the prediction respectively, $\beta^i$ is the weight of the $i_{th}$ class. Here, the weights $\beta^i$ are set to 1.0, 2.0 ,4.0 and 4.0 for kidney, tumor, cyst and background respectively, in Eq. 4 according to the preliminary experiments.

## 3   Experiments

In this section, we illustrate the KiTS21 dataset on Section 3.1. Then, the evaluation metrics are presented on Section 3.2. Next, we describe the pre-process and post-process methods on Section 3.3. Finally, we specify the implementation details on Section 3.4.

### 3.1   Datasets

The CT scans used in this work come from The 2021 Kidney and Kidney Tumor Segmentation Challenge(KiTS21), which contains 300 complete data of kidneys, kidney tumors, kidney cysts and background labels. We randomly divided 60 data into one big category, divided into five in total, for five-fold cross-validation.

The data labels provided by the KiTS21 challenge are not exactly the same. Some data contains 4 complete labels, and some may only contain 3 of them. This also makes the training of our model difficult. Fig.3 shows some of them.

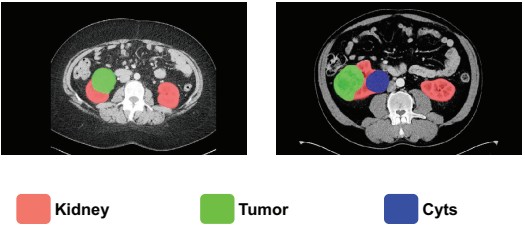

**Fig. 3.** Illustration of the annotations in the KiTS21 dataset.

### 3.2   Metrics

This article employs Dice and surface Dice similarity coefficient (surface DSC)[10] to evaluate our segmentation results. Dice is one of the most commonly used evaluation indicators. Specifically, we use kidneys, kidney tumors, and cysts as the foreground, everything else as background to calculate Dice scores. In medical images, the voxel spacing is usually unequal, and the calculation of surface voxels usually leads to larger errors. The surface Dice used in this article is given an allowable error distance, and the surface within this error range is regarded as the overlapping part, the surface overlap dice value of the ground truth mask and the predict mask is calculated.

### 3.3   Pre- and post-processing

We consider using 2D U-Net to complete our experiments. KiTS21 challenge provides 3D dataset, we first convert voxels into slice data, highlight organ and tumor features through threshold processing and threshold normalization, enhance the data by flipping, random cropping, and random translation.

In the post-processing part, considering the influence of noise, we perform the operation of the largest connected domain on the data to eliminate the noise.

### 3.4   Implementation details

First, we initialize the model parameters, set the training epoch to 20, the initial learning rate to $10^{-5}$, batchsize is set to 8, and use the Adam optimizer. The proposed network was implemented in python using Pytorch(v1.5.1) framework in the backend. All training and testing experiments are run on a workstation with an NVIDIA GeForce GTX 2080Ti with 11G GPU memory.

## 4   Result

We evaluated our model on KiTS21 dataset through five-fold cross-validation. The final results were obtained by averaging the best performance of each fold.

Table 1 reports the quantitative results of the proposed SeResUNet and the most commonly used segmentation methods. The dice of kidney, masses and tumor are 91.60%, 58.80%, 54.16% respectively, and the surface dice are 84.62%, 37.91%, 37.59% respectively. In addition, the segmentation results are shown in Fig.4.

**Table 1.** Dice score (mean) and Surface Dice of the proposed method on 5-fold Cross Validation.

| Method | Kidney (Dice) | Masses (Dice) | Tumor (Dice) | Kidney (SD) | Masses (SD) | Tumor (SD) |
|---|---|---|---|---|---|---|
| 2D U-Net | 0.9132 | 0.3769 | 0.3712 | 0.8425 | 0.2618 | 0.2573 |
| SeResUNet18 | 0.8801 | 0.3923 | 0.3556 | 0.7952 | 0.2677 | 0.2370 |
| SeResUNet50+D | 0.9144 | 0.5797 | 0.5274 | 0.8396 | 0.4187 | 0.3741 |
| SeResUNet18+D | 0.9160 | 0.5880 | 0.5416 | 0.8462 | 0.3791 | 0.3759 |

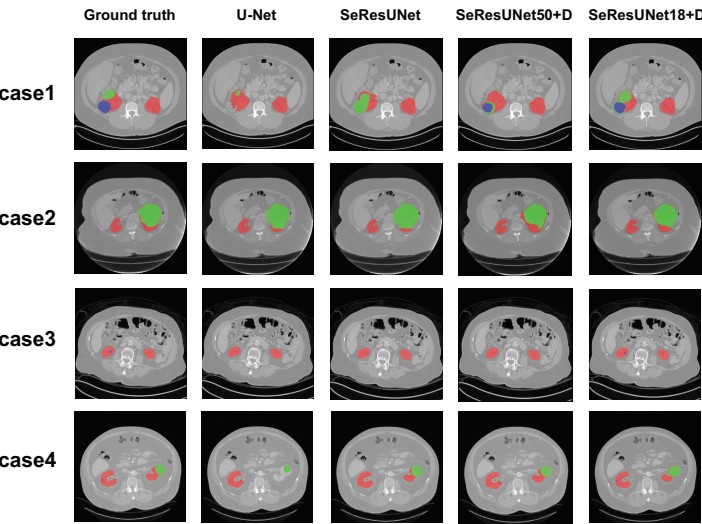

**Fig. 4.** Qualitative comparison of segmentation results for KiTS21 dataset test:Ground truth, U-Net, SeResUNet, SeResUNet50 with deep supervision(SeResUNet50+D), and proposed SeResUNet18 with deep supervision(SeResUNet18+D). Color coding: red,Kidney; green,Tumor; blue,Cyts.

## 5    Discussion and Conclusion

In this work, we proposed a novel segmentation network called SeResUNet to deal with the kidney and tumor segmentation task. First, we adopt the encoder-decoder architecture like U-Net, and use ResNet to deepen the network in the

encoder. At the same time, in order to avoid deep network degradation problems and speed up the convergence, we add multi-level deep supervision to the decoder. In addition, we noticed the importance of different channels and introduced Squeeze-and-Excitation module, which automatically obtains the weight of each feature channel through learning, and then highlights useful features based on this weight and suppresses features that are not useful for the current task. Finally, the weight cross-entropy loss function is used to solve the problem of data imbalance. Through the evaluation on the KiTS21 dataset, it can be seen that the model we proposed has a stronger ability in the kidney and its tumor segmentation.

It can be seen from table 1 that our network is improved by 0.28%, 21.1% and 16.88% respectively compared with the classic network 2D U-Net. The segmentation results of the kidney have not improved much, but the segmentation results of tumors and cysts have improved greatly, indicating that our model performs better in the subtle parts. Compared with Se-ResUNet18 and Se-ResUNet50, kidney, masses and tumor are increased by 0.16%, 0.83%, 1.42% respectively, it shows that the results obtained by deeper networks are not necessarily better. In addition, we also compare whether to use deep supervision. The experimental result shows that the segmentation results of our model after adding deep supervision will be much better.

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
