# OpenReview forum: "Squeeze-and-Excitation Encoder-Decoder Network for Kidney and Kidney Tumor Segmentation in CT images"
_MICCAI.org/2021/Challenge/KiTS — Submitted to KiTS21 Challenge_

### Official Review · Reviewer_yyY8 · 2021-08-30

**Rating:** 8

**Review:**

The authors provide a nice in-depth paper describing their approach. They make effective use of figures and tables and do a great job including fine-grained details about their networks and experiments. One crucial detail that is missing, however, is the method with which the authors aggregated instance segmentations into composite segmentations that could be used for training and validation. Did you use majority voting? If so, you should explicitly say so.

---

### Official Review · Reviewer_iQAp · 2021-08-30

**Rating:** 7

**Review:**

### Introduction

- Framwork -> framework
- segmentationwhich -> segmentation which

### Methods

- Looks good

### Results

- Looks good

### Discussion and Conclusion

- You say that " The experimental result shows that the segmentation results of our model after adding deep supervision will be much better.". This makes it seem like you have not yet implemented deep supervision. Is this the case? Your results section makes it seem like you have already implemented this and are reporting the results.

---

### Decision · Program_Chairs · 2021-08-30

**Decision:**

Minor Revisions

**Comment:**

Please address the reviewer comments and resubmit